



# Rhizodeposition efficiency of pearl millet genotypes assessed on short growing period by carbon isotopes ($\delta^{13}$C and F$^{14}$C)

Papa Mamadou Sitor Ndour[1,t], Christine Hatté[2], Wafa Achouak[3], Thierry Heulin[3] and Laurent Cournac[1]

[1] Eco&Sols, Université de Montpellier-IRD-CIRAD-INRAE-Institut Agro, Montpellier, France
[2] LSCE, CEA-CNRS-UVSQ-Université Paris-Saclay, 91191 Gif-sur-Yvette Cedex
[3] LEMiRE-BIAM, Aix Marseille University-CEA-CNRS, FR ECCOREV 3098, F-13108 Saint-Paul-Lez-Durance, France
[t] Current address: PMI-Laboratory, Agrobiosciences, Mohammed VI Polytechnic University, Ben-Guerir, Morocco

*Correspondence to*: Papa Mamadou Sitor NDOUR (sitndour@yahoo.fr)

**Abstract.** Rhizosheath size varies significantly with crop genotype, and root exudation is one among its driving factors. Unravelling the relationships between rhizosheath formation, root exudation and soil carbon dynamics may bring interesting perspectives in terms of crop breeding towards sustainable agriculture. Here we grew four pearl millet (C$_4$ plant type: $\delta^{13}$C of -12.8 ‰, F$^{14}$C = 1.012) inbred lines showing contrasting rhizosheath sizes in a C$_3$ soil type (organic matter with $\delta^{13}$C of -22.3 ‰, F$^{14}$C =1.045). We sampled the root-adhering soil (RAS) and bulk soil after 28 days of growth under semi controlled
condition. The Soil organic carbon (SOC) content, $\delta^{13}$C and F$^{14}$C of soil samples were measured, and the plant-derived C amount and $C_{lost}/C_{new}$ ratio in RAS were calculated. The results showed a significant increase in $\delta^{13}$C in the RAS of the four pearl millet lines compared to the control soil, suggesting that this approach was able to detect plant C input to the soil at early stage of pearl millet growth. The concentration of plant-derived C in RAS did not vary significantly between pearl millet lines, but the absolute amount of plant-derived C varied significantly when we considered the RAS mass of these different lines.
Using a conceptual model and data from the two carbon isotopes measurements, we evidenced a priming effect for all pearl millet lines. Importantly, the priming effect amplitude was more important for the low-aggregation lines than for the high-aggregation ones indicating a better C sequestration potential of these latter.

**Keywords: *Root exudation; Rhizosphere; Priming effect; Root-adhering soil; SOC; Pearl millet***

## 1. Introduction

In the context of climate change, a new challenge for agriculture is to sequester more carbon in the soil to mitigate CO$_2$ increase in the atmosphere (Lal et al., 2015). This will be particularly important as increasing SOC content would have beneficial effects on agriculture by increasing soil fertility, and then improving food security (Lal et al., 2015). Moreover, this strategy could be particularly relevant in the Sahel region of Africa, where very little above-ground cover remains after harvest, leading to soil carbon depletion (Baudron et al., 2014).



Pearl millet (*Pennisetum glaucum* L.R.Br ) is the main cereal grown in the Sahel and the arid region of India (Debieu et al., 2017). It is naturally adapted to drought conditions of these semi-arid and arid regions and is a staple food for nearly 100 million people around the world (Burgarella et al., 2018; Varshney et al., 2017) and 50 million in West Africa. Following rainfall decline in recent decades, farmers have privileged short-cycle varieties that give optimal yield to cope with the short rainy season (Vigouroux et al., 2011). Despite this adaptation, pearl millet yields remain very low (less than 1 ton/hectare)

compared to other cereals. The main factors explaining this low productivity are (i) low soil fertility associated with low soil organic carbon (SOC) content due in part to the export of crop residue and low carbon fertilization, and (ii) climatic uncertainties (mainly rainfall) which could increase with climate change.

    To simultaneously improve soil carbon sequestration through crop cultivation and ensure food security in Africa, a combination of plant breeding and the development of improved agricultural practices could be considered. Then, plant

breeding for root traits (architecture and root exudation) could be an original strategy to improve water and mineral nutrition of crops and to enhance SOC sequestration. Root exudation is reported to account for 17 to 40 % of plant photo-assimilates (Nguyen, 2003; Badri and Vivanco, 2009; Sasse et al., 2018). However, the question of whether cultivar variability can be used to significantly increase plant carbon deposition and its sequestration in soil is not fully documented. We have shown in a previous study that the mass of root-adhering soil (as an estimator of rhizosheath) varied significantly with pearl millet

genotype among a panel of inbred lines (Ndour et al., 2021). This new functional trait for plant phenotyping is linked to root exudation and other functional and morphological root traits such as root architecture (Delhaize et al., 2012, 2015; George et al., 2014) and to root-associated microbiota (Ndour et al., 2020). Therefore, the carbon input of pearl millet into the soil must be directly determined to achieve the carbon sequestration objective. However, due to the heterogeneous nature of the soil and particularly in field conditions, conventional carbon measurement methods could not answer this question in short term

experiments used for screening plant genotypes. The change in carbon concentration would remain below the variability around the mean value. Furthermore, it is now well known that the input of energetic molecules (such as roots exudate) induces an increase in the activity of microorganism and can thus contribute to extra mineralization of molecules derived from soil old carbon-, that is the so-called priming effect (Fontaine and Barot, 2005) Assessing the durability of new carbon storage, *i.e.,* assessing a balance between carbon gain and loss, cannot be achieved by looking only at carbon concentration on a point-by-

point basis. The persistence of the carbon injected into the soil must be monitored over several years. As an alternative method, measuring carbon deposition in the rhizosphere using carbon isotopes ($^{13}$C and $^{14}$C) that are much less sensitive to soil heterogeneity could be very interesting to test and indicative of the age of the primed carbon. For instance, based on the fact that C$_4$ plant organic matter is enriched in $^{13}$C compared to C$_3$ plant organic matter, He et al. (2019) used carbon isotopic shift ($\delta^{13}$C) in a wetland with initial C$_3$ soil type colonized by a C$_4$ plant (*Spartina alterniflora*) in order to determine SOC deposition

and its carbon sequestration potential. Likewise, analysis of soil $^{14}$C profiles has revealed the ability of soil to store carbon and the impact of certain practices on both carbon stock and its persistence (Mathieu et al., 2015; Jreich et al., 2018).

    The objective of this work was to apply these alternative methods to measure carbon deposition in the rhizosphere of pearl millet using carbon isotopes ($^{13}$C and $^{14}$C) and to compare the ability of different pearl millet inbred lines to durably



deposit carbon in the soil. We used four pearl millet lines with contrasting phenotype for root-adhering soil aggregation (*i.e.*
rhizosheath size) estimated by RAS/RT ratio (root-adhering soil mass/root tissue mass).

## 2. Material and methods

### 2.1 Pearl millet cultivation

Four pearl millet lines with contrasting RAS/RT ratios (*ie* rhizosheath size or rhizosphere aggregation) were selected from a
previous study (Ndour et al., 2021): one low-aggregation line: L220, one intermediary-aggregation line: L3, and two high-
aggregation lines: L253 and L132. The soil was sampled at Fissel Mbedap, Thiès, Senegal (14°29'21.8 N 16°35'36.4 W) in an
uncultivated area essentially populated by *Guiera senegalensis* ($C_3$ plant species). The soil was sampled at the surface horizon
(0-20 cm) and sieved at 2 mm and homogenized. The $\delta^{13}C$ of the soil was -22.3 ‰. The experiment was performed in
greenhouse under natural light condition on August 2018. The soil was filled in WM pots installed in plastic crates (Ndour et
al., 2021). Each pot was filled with 1.5 kg of the soil moistened to its water holding capacity. The four pearl millet lines were
sown in a randomized bloc design with 7 replicates. Seven pots without plant were added to serve as controls. After one week,
we thinned to one plant per pot. The seedlings were watered daily with 20 mL throughout the experiment until one day before
the soil sampling.

### 2.2 Soil sampling

Harvesting was performed 28 days after sowing by detaching the two pieces of the pots and shaking gently to keep only the
soil adhering closely to the roots (RAS). Roots and RAS were separated by washing the roots in 50 mL. We also sampled bulk
soil for each pot (away from the root system) as well as control soil from unplanted pots. Soil samples were dried by incubation
at 65 °C for 4 days. All soil samples were stored in haemolysis tubes until laboratory analyses. Shoot and root biomass were
weighted and shoot biomass was also sampled for $\delta^{13}C$ measurement of pearl millet tissue.

### 2.3 Laboratory analyses

#### 2.3.1 Organic carbon content

Approximately 15 to 20 mg of soil samples were weighed in tin cups for measurement (with a precision of 1 to 2 µg). The
sample was combusted in a FlashEA1112, and the carbon content determined using the Eager software. One standard was
inserted every 10 samples. For the range of carbon content of the soils analysed here, the relative measurement error is of the
order of 1.5-2%.



### 2.3.2 δ $^{13}$C analysis

Isotopic analysis was performed online using a continuous flow EA-IRMS coupling, i.e. a FlashEA1112 Elemental Analyzer coupled to a Thermo Finnigan Delta+XP Isotope-Ratio Mass Spectrometer. Three in-house standards (Hobo5 sediment: $\delta^{13}$C = -13.4‰, oxalic acid: $\delta^{13}$C = -16.7‰ and GCL with $\delta^{13}$C = -26.7‰) were inserted every five samples. Each in-house standard was regularly checked against international standards. The results are reported in the δ notation:

$\delta^{13}$C = ($R_{sample}/R_{standard} - 1$), where $R_{sample}$ and $R_{standard}$ are the $^{13}$C/$^{12}$C ratios of the sample and the international standard, Vienna Pee Dee Belemnite (VPDB), respectively. The measurements were triplicated for representativeness. The external reproducibility of the analysis was better than 0.1‰, typically 0.06‰.

### 2.3.3 $^{14}$C Geochronology

The amount of soil needed to yield 1 mg of carbon was weighted into a tin capsule. Soil carbon was successively converted into $CO_2$ and reduced in C in presence of $H_2$, using an Automated Graphitization Equipment, AGE3 (Wacker et al., 2010). The pure graphite is then pressed in the presence of ultrapure iron into a target to be introduced in the solid source of *ECHo*MICADAS (Synal et al., 2007; Tisnérat-Laborde et al., 2015), a Compact Radiocarbon System able to run very small samples. Results were reported as F$^{14}$C as recommended by Reimer et al. (2004).

### 2.4 Calculations and statistical analyses

The proportion of plant-derived carbon (*p*) in the root-adhering soil was calculated according to the following equation (1):

$$p = \frac{\delta13C_{RAS} - \delta13C_{control}}{\delta13C_{mil} - \delta13C_{control}} \quad (1)$$

Where $\delta_{RAS}$ represents the $\delta^{13}$C of the root-adhering soil organic carbon, $\delta_{control}$ represents the $\delta^{13}$C of the unplanted soil organic carbon, and $\delta_{mil}$ represents the $\delta^{13}$C of pearl millet tissues we measured (-12.8 ‰).

The plant-derived carbon content (*PDCC$_{RAS}$*) of the root-adhering soil was calculated by multiplying the proportion of plant derived carbon (*p*) by the soil organic carbon (SOC) content:

*PDCC$_{RAS}$* (gC. kg$^{-1}$ soil) = *p* x SOC (gC. kg$^{-1}$ soil)

The plant-derived carbon deposited in the root-adhering soil volume (*PDCD$_{RAS}$*) was calculated by multiplying the *PDCC* by the average root-adhering soil mass (RAS) of the different pearl millet lines in **Table A1**. This is expressed in a per plant basis:

*PDCD$_{RAS}$* (gC) = *PDCC$_{RAS}$* (gC. kg$^{-1}$ soil) x mass of RAS (kg)

The Plant-derived C deposited/Plant biomass was expressed in (%):

Plant-derived C/Plant biomass (%) = *PDCD$_{RAS}$* / Plant biomass x100

The normality of the data was tested using Shapiro's test and then ANOVAs and the Tukey post-hoc test ($p < 0.05$) were performed to compare the mean of the different parameters.



## 2.5 Processed data: integrated model

Three equations that derive from our analyses help in conceptualizing soil carbon dynamics:

$$C_{tot} = C_{old} + C_{new} - C_{lost} \quad (2)$$

$$C_{tot}\, \delta^{13}C_{tot} = C_{old}\, \delta^{13}C_{old} + C_{new}\, \delta^{13}C_{new} - C_{lost}\, \delta^{13}C_{lost} \quad (3)$$

$$C_{tot}\, F^{14}C_{tot} = C_{old}\, F^{14}C_{old} + C_{new}\, F^{14}C_{new} - C_{lost}\, F^{14}C_{lost} \quad (4)$$

with subscript "tot" for total organic carbon after experiment, "old" for organic carbon before the experiment, "new" for the new carbon input during the experiment and "lost" for organic carbon that was present before the experiment and was lost during the experiment. "C" is for the carbon concentration.

Three hypotheses were investigated:

H1        input of new carbon without change in original carbon cortege $C_{lost}=0$

H2        steady state: replacement of old carbon by new carbon        $C_{lost}=C_{new}$

H3        priming effect: replacement and loss of old carbon        $C_{lost} > C_{new}$

The isotopic information for "old" carbon was retrieved from the control soil. The "new" carbon input was derived from millet rhizodeposition. As first approximation, the new C in the RAS fractions ($\delta^{13}C_{new}$) was considered as equal to millet leaf $\delta^{13}C$. As first approximation, $\delta^{13}C_{lost}$ was considered as equal to $\delta^{13}C_{old}$. Solving the equation provided us with information on newly input carbon ($C_{new}$) and released carbon ($C_{lost}$, $F^{14}C_{lost}$). The derived parameters have been gathered in Table 2 in the Result section.

## 3. Results

### 3.1 SOC, $\delta^{13}C$ and $F^{14}C$ of soil samples

Soil organic carbon (SOC) content was not significantly different between RAS, bulk soil and unplanted control fractions in any of the pearl millet lines, which was expected given the method of measuring C (combustion) and the short duration of the experiment (28 days) **(Fig. 1A)**. Using natural $^{13}C$ abundances ($\delta^{13}C$), significant increases were noticed in the root-adhering soil fraction ($\delta^{13}C_{RAS}$) of all four pearl millet lines compared to the unplanted control soil ($p <0.05$), in agreement with the higher $\delta^{13}C$ value of pearl millet leaves and probably also root exudates (-12.8‰) compared to that of soil with C3 plant history (-22.3‰). On the contrary there is no significant difference between all four bulk soils and the control soil showing that there is no significant input of fresh C4-derived carbon outside the rhizosphere. For all four pearl millet lines, the $\delta^{13}C$ of the RAS fractions was always higher than that of the bulk soil, with the difference being significant for only two lines (L3 and L253). No significant difference was found in the $\delta^{13}C$ measured in the RAS between the four pearl millet lines **(Fig. 1B)**.





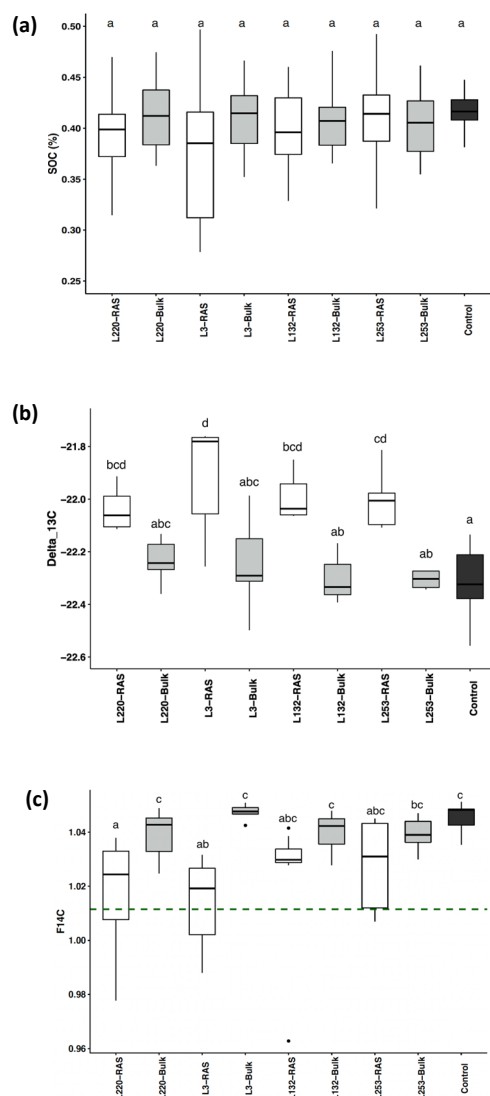

**Figure 1**: (A): Soil organic carbon (SOC) content of the RAS (Root-adhering soil), Bulk soil and Control unplanted soil. (B) $\delta^{13}C$ in the RAS, Bulk soil and Control soil. (C) $F^{14}C$ in the RAS and Bulk soil of pearl millet lines and for the Control soil. The dotted line indicates the $F^{14}C$ value of pearl millet tissue, i.e. the new carbon $F^{14}C$ ($F^{14}C=1.012 \pm 0.002$). Different letters indicate significant difference using Tukey post-hoc test ($p < 0.05$).





The plant-derived carbon content of the RAS varied from 0.124 gC. kg$^{-1}$ soil (L220) to 0.192 gC. kg$^{-1}$ soil (L3), but
no significant difference was evidenced between the four pearl millet lines (**Fig. 2A**). Considering the mass of root-adhering
soil of the four pearl millet lines (Table S1), the amount of plant-derived carbon deposited in the RAS varied significantly
according to pearl millet line and was significantly higher for the high-aggregation line (L253) compared to low-aggregation
line (L220) and intermediary-aggregation one (L3) **(Fig. 2B).** The same trend was found for carbon deposited in the root-
adhering soil expressed by plant total biomass (**Fig. 2C**).

The F$^{14}$C values of the bulk soil of the four pearl millet lines were not significantly different from those of the
unplanted soil control (**Fig. 1C**), showing the absence of $^{14}$C dilution away from the rhizosphere. The low-aggregation line
(L220) and intermediary-aggregation line (L3) significantly diluted the $^{14}$C abundance in the RAS fraction compared to the
bulk soil, in contrast to the two high-aggregation lines (L132 and L253) characterized by F$^{14}$C values in RAS non-significantly
different from those of the bulk soil (**Fig. 1C**).

The F$^{14}$C values of bulk soil and unplanted soil control were significantly higher (F$^{14}$C > 1.04) than those of pearl
millet leaves (F$^{14}$C= 1.012 ± 0.002, dotted line in **Fig. 1C**) illustrating a significant incorporation of carbon contemporaneous
of the bomb peak into the 'old' soil carbon pool. The dilution of the F$^{14}$C in root-adhering soil fractions, due to the root
exudation of C compounds with the same F$^{14}$C value as leaves was significantly different for the two high-aggregation lines
(L253 and L132) compared to the two other lines (L220 and L3) (**Fig. 1C.**)



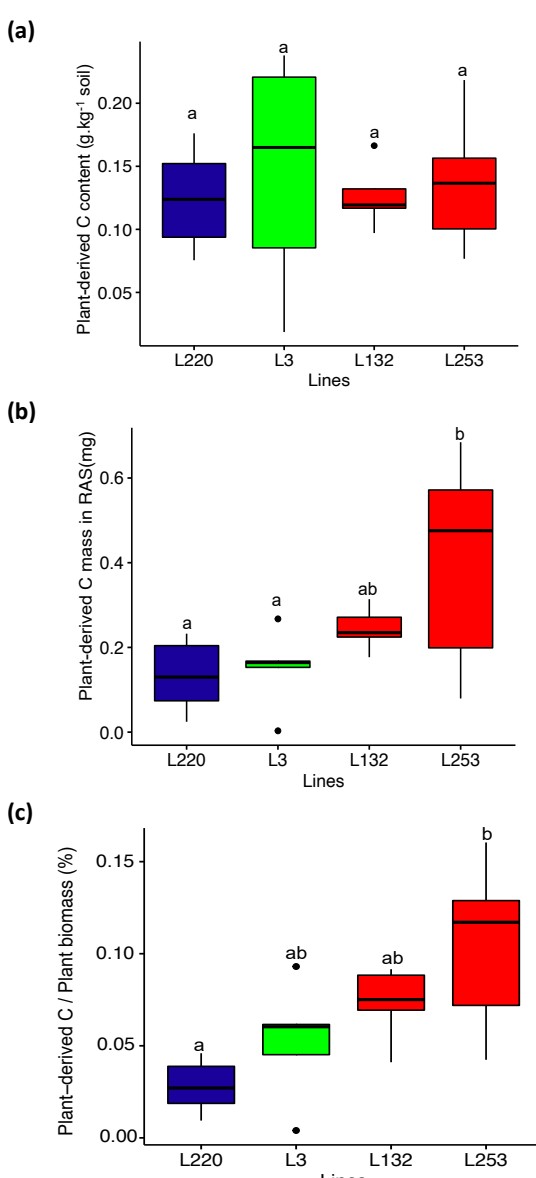

**Figure 2**: (A) Plant derived-carbon content (PDCC in gC/kg of soil) in the root-adhering soil (RAS) of the four pearl millet lines. (B) Plant derived-carbon mass deposed (PDCD in mg C) in the RAS of the four pearl millet lines. (C) Plant-derived carbon amount per plant biomass (in %) produced by the four pearl millet lines. Different letters indicate significant difference using an ANOVA and the Tukey post-hoc test ($p < 0.05$).





## 3.2 Conceptual modelling

Solving equations (2), (3), (4) did not provide a solution under either hypothesis 1 (input of new carbon without change in original carbon cortege) or hypothesis 2 (steady-state). Only hypothesis 3 made it possible to solve the equations and provided consistent parameters (**Table 1**). The validation of this hypothesis showed that the experiment (4 weeks of pearl millet growth)

is in a priming effect phase. For all pearl millet lines, new carbon has been deposited, and some of the 'old' carbon released. At this stage of plant growth, the trends showed a higher carbon loss vs gain (high $C_{lost}/C_{new}$ ratio) for the low-aggregation and intermediary-aggregation lines L220 and L3 (4.4 and 3.5) compared to the two high-aggregation lines L132 and L253 (2.9 and 3.3) (**Table 1**). Investigation with $^{14}C$ isotope allowed the lost carbon to be specified. The $F^{14}C$ value of lost carbon was between 1.2 and 1.3, *i.e* carbon which is consistent with the 1980s (Hua et al., 2013). This means that the injection of new

carbon implies also loss of some decades old carbon.





**Table 1**: data processing for the root-adhering soil (RAS) of pearl millet lines. Hypotheses 1 and 2 are not solvable with the measured parameters. Hypothesis 3 is the only one that has a solution. The amount of carbon added ($C_{new}$), the amount of carbon lost ($C_{lost}$) and the radiocarbon fraction of this lost carbon ($F^{14}C_{lost}$) are then evaluated. Carbon content is expressed as %wt, $\delta^{13}C$ as ‰, $F^{14}C$ is unitless. The values are given with a digit number in accordance with the number of significant figures.

| | | L220 | L3 | L132 | L253 |
|---|---|---|---|---|---|
| **Hypothesis 1: $C_{lost} = 0$** | | | | | |
| *RAS* | $C_{lost}$ | 0 | 0 | 0 | 0 |
| *Control* | $C_{old}$ | 0.43 | 0.43 | 0.43 | 0.43 |
| $C_{tot}$-$C_{old}$ | $C_{new}$ | *no solution* | *no solution* | *no solution* | *no solution* |
| **Hypothesis 2: $C_{lost} = C_{new}$** | | | | | |
| *Control* | $C_{old}$ | 0.43 | 0.43 | 0.43 | 0.43 |
| *RAS* | $C_{tot}$ | 0.39 | 0.38 | 0.40 | 0.40 |
| $C_{tot}=C_{old}$ | | *not fulfilled* | *not fulfilled* | *not fulfilled* | *not fulfilled* |
| **Hypothesis 3: $C_{lost} - C_{new} > 0$** | | | | | |
| *Control* | $C_{old}$ | 0.43 | 0.43 | 0.43 | 0.43 |
| *RAS* | $C_{tot}$ | 0.39 | 0.38 | 0.40 | 0.40 |
| | $\delta^{13}C_{old}$ | -22.3 | -22.3 | -22.3 | -22.3 |
| | $\delta^{13}C_{tot}$ | -22.1 | -21.8 | -22.0 | -22.0 |
| | $\delta^{13}C_{new}$ | -12.8 | -12.8 | -12.8 | -12.8 |
| | $\delta^{13}C_{lost}$ | -22.3 | -22.3 | -22.3 | -22.3 |
| *(2)+(3) ->* | $C_{lost}$ | 0.040 | 0.066 | 0.038 | 0.039 |
| *(2) ->* | $C_{new}$ | 0.009 | 0.019 | 0.013 | 0.012 |
| | $C_{lost}/C_{new}$ | 4.4 | 3.5 | 2.9 | 3.3 |
| | $F^{14}C_{old}$ | 1.045 | 1.045 | 1.045 | 1.045 |
| | $F^{14}C_{tot}$ | 1.017 | 1.014 | 1.024 | 1.028 |
| | $F^{14}C_{new}$ | 1.012 | 1.012 | 1.012 | 1.012 |
| *(4) ->* | $F^{14}C_{lost}$ | 1.311 | 1.212 | 1.257 | 1.213 |



## 4. Discussion

Our data showed no statistically significant changes in soil carbon content between the RAS of the different pearl millet lines,
neither with the bulk soil nor with the control soil, after 28 days of growth. Similar results were obtained in other studies in
which no significant differences were noticed in the rhizosphere SOC content compared to control soil (Wang et al., 2016;
Van de Broek et al., 2020). In contrast, using natural $^{13}C$ abundances we demonstrated a significant increase in the $\delta^{13}C$ of the
root-adhering soil fraction of the four pearl millet lines compared to their respective bulk soil fractions and to the unplanted
control soil ($p < 0.05$). This shift in $\delta^{13}C$ is in agreement with the higher $\delta^{13}C$ value of pearl millet leaves (-12.8‰) and probably
also of root exudates compared to that of soil with a $C_3$ plant history (-22.3‰), evidencing the carbon deposition into the soil
through root exudation. Nevertheless, plant-derived C content (gC/kg soil) in the rhizosphere (RAS) did not vary significantly
between pearl millet lines. The trend for increasing aggregation efficiency between pearl millet lines was not found in the $\delta^{13}C$-
derived evaluation of input carbon content. This result is consistent with those obtained by Broek et al., (2020) who found no
significant differences in net carbon rhizodeposition between four wheat cultivars with contrasting root biomass (including
two old and two recent varieties). However, the absolute amount of plant-derived C in RAS (obtained by multiplying plant-
derived C content by the mean RAS mass of the different pearl millet lines) was significantly higher for the high-aggregation
pearl millet line (L253) compared to the low- and intermediary-aggregation lines (L220 and L3, respectively). The ratio
between this plant-derived C deposited in the RAS and total plant biomass (expressed in %) was also higher for the high-
aggregation pearl millet line (L253) compared to the L220 and L3 lines, allowing for the detection of about 0.1% of plant-
derived C per unit of plant biomass. This indicates that the significant difference in root exudation among these pearl millet
lines would be related to a difference in carbon allocation rather than a difference in biomass quantity production.

The distinct evolution between the $F^{14}C$ contents of the RAS of the different pearl millet lines and the bulk soil clearly
showed a different distribution of soil carbon age between the two fractions. The decrease in $F^{14}C$ in the RAS fraction of the
L220 and L3 lines may result from an increase in the proportion of 'new' carbon from root exudation with $F^{14}C$ equal to that
of the millet leaf (see Figure 12 in Jreich et al. 2018), compared to that of the bulk soil (equivalent to control soil). This would
be a dilution effect by incorporation of 'new' carbon. Here we are probably faced with contrasting situations: a loss of 'old'
carbon due to the priming effect of root exudates with a partial replacement of the lost 'old" carbon by 'new carbon' from root
exudates and probably their associated microbial metabolites, this replacement tending to be of lower amplitude in low
aggregation lines (case of L220 and L3) than in high aggregation lines (case of L253 and L132). The $^{14}C$ isotope analyses
allow us to estimate both the ratio of carbon input to output and the mean age of the lost carbon, which is a few decades
(1980s).

Mwafulirwa et al. (2016) showed a genotypic influence of barley on the stabilization of rhizodeposited C and the
SOM mineralization. Using a combination of natural isotope tracing and a conceptual modelling, we confirmed these findings

in pearl millet as our results show that the pearl millet lines L253 and L132 better preserve the soil existing C stock ('old' C)
after plant deposition of 'new' C than the lines L220 and L3. In addition, the better C preservation (lower priming) was
importantly noticed with the high-aggregation genotypes (L132 and L253). This suggest that C preservation of these pearl
millet lines could be related to its transformation into microbial macromolecules such as exopolysaccharides which make C
more stable and contribute to increase rhizosphere soil aggregation as reported in previous studies (Gouzou et al., 1993; Alami
et al., 2000; Bezzate et al., 2000).

## 5. Conclusion

These results show that natural carbon isotope tracing allowed us to detect pearl millet root exudation into soil at early stage
of plant growth ($\delta^{13}$C; at 28 days) and to study the dynamics of 'old' and 'new' C, demonstrating the impact of the rhizosphere
priming effect at this stage of plant growth ($F^{14}$C). With the use of a conceptual modelling applied to data from these two
isotopes, we pinpoint a way of assessing how root C metabolism of some plant lines/varieties can contribute effectively to the
storage of carbon in the soils with an enrichment in- or a better preservation of- SOM, and therefore contribute to a reduction
of greenhouse gases balance in agriculture.

**Data availability**

The data generated in this study are available from the corresponding authors upon reasonable request

**Author contributions**

PMSN, CH, WA, TH and LC conceptualized the study. PMSN performed the experiment and the soil sampling. CH performed
laboratory analyses and the conceptual modelling. PMSN wrote the initial draft. PMSN, CH, WA, TH and LC contributed to
generating and reviewing the subsequent versions of the manuscript.

**Competing interests**

The authors declare that they have no conflict of interest

**Acknowlegments**

PMS Ndour wish to acknowledge the French National Research Institute for Sustainable Development (IRD) and the Make
Our Planet Great Again (MOPGA) initiative of the French government for co-funding his postdoctoral fellowship.



**Financial support**

This work benefited from the support of the Agence Nationale de la Recherche (ANR) through the RootAdapt grant (N° ANR-250 17-CE20-0022) and the CEA (CarbonSeq DRF impulsion grant).

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

**Appendix A:**

**Table A1:** RAS/RT means (Rhizosheath size) of the four pearl millet lines measured in Ndour et al (2021); Root (RT) mass (g), shoot mass (g), total plant biomass (g) measured for the four pearl millet lines; and the mean values of the Root-adhering soil (RAS) mass (g) calculated for the four pearl millet lines. All these parameters are measured/calculated at 28 days of growth.

| Lines | RAS/RT | Root (RT) (g) | Shoot (g) | Plant Biomass (g) | RAS (g) |
|-------|--------|---------------|-----------|-------------------|---------|
| L220 | 8.4 | 0.116 (± 0.07) | 0.286 (± 0.08) | 0.402 (± 0.15) | 0.972 (± 0.6) |
| L3 | 14.9 | 0.096 (± 0.06) | 0.194 (± 0.07) | 0.290 (± 0.13) | 1.433 (± 0.9) |
| L132 | 23.3 | 0.082 (± 0.04) | 0.231 (± 0.03) | 0.313 (± 0.06) | 1.921 (± 0.8) |
| L253 | 20.8 | 0.128 (± 0.06) | 0.244 (± 0.07) | 0.372 (± 0.13) | 2.659 (± 1.2) |
