# Peer review of "Rhizodeposition efficiency of pearl millet genotypes assessed on short growing period by carbon isotopes ( $\delta^{13}$ C and $F^{14}$ C)"

_SOIL, 2021_

## Author Comment (AC2)

**Manuscript number: Soil-2021-108**

Title: Rhizodeposition efficiency of pearl millet genotypes assessed on short growing period by carbon isotopes (δ13C and F14C)

**#Topical Editor comment:**

**General comment**: One thorough review was obtained for your paper and unfortunately another reviewer withdrew. I have read the paper as well to provide some additional comments. In general, little is needed before this paper can be accepted for publication. Please be careful on the use of the term 'rhizosphere' when conducting measurements of rhizosheath soil. This just needs terms changed in a few places. Details of 'water holding capacity' are needed in section 2.1. Mention value and the approach used to get this measurement. Also give brief details on how the soil was moistened.

> *Response: We thank you for handling our submission, and for serving as a second referee with the withdrawal of the reviewer 2.*
>
> *We have provided in the "**Pearl millet cultivation**" subsection the value of the water holding capacity and detail of its determination: " Each pot was filled with 1.5 kg of the soil moistened to its water holding capacity (8%) which was determined gravimetrically from the graph of water loss kinetics during 18 hours. This watering level was adopted to have a good germination rate as the soil was sandy with a low water retention capacity, and we have used bottomless pots which allow water infiltration as in Ndour et al (2017)"*

**Section 2.2.3.**

**Comment 1**: line 99 - 'weighed'

> *Response 1: the suggested correction was done*

**Comment 2:** line 100 - 'in the pressence of…'

> *Response 2: the suggested correction was done*

**Comment 3:** Line 101 'graphite was then...'

> *Response 3: This suggestion was adopted*

**Section 3**

**Comment 4**: Line 145 - 'rhizosheath' rather rhizosphere

> *Response 4: This suggestion was adopted*

**Comment 5**: Figure 1 - put units on the SOC % plot so i is clear it is gravimetric and not volumetric

> *Response 5: This suggestion was adopted, the unit of SOC is expressed in (%wt) and this was mentioned on the Y axe title of **Fig. 1 (a)***

**Comment 6**: line 155 - 'was found between...'

*Response 6: We replace "evidenced" by "found"*

**Comment 7**: line 158 - '(L220) and intermediary-aggregation (L3) lines...'

**Response 7**: *This suggestion was adopted*

**Comment 8**: Fig 2 - be consistent with units. (a) and (b) should both be g kg$^{-1}$

*Response 8: In figure 2 (b) we expressed Plant-derived C mass in (g). We think the absolute value is more informative than a relative value, here.*

**Comment 9**: Fig. 2 caption - 'deposited' rather than 'deposed'

*Response 9: This suggestion was adopted*

**Comment 10**: Line 178 - '..hypothesis 3 (priming)'

*Response 10: We adopted this suggestion: only hypothesis 3 (**priming**) made it possible to solve the equations*

**Comment 11**: Future research needs/outlook is missing at present. Secondary impacts to improved soil properties would also benefit from being described

*Response 11: We added the research perspectives in the conclusion: " Further studies should focus on studying the profile of root exudates among these pearl millet genotypes to see in what extent it could contribute to the variation observed in plant-derived C dynamic in the rhizosphere. Nevertheless, whatever the relative contribution of the quantity and the quality of root exudation in shaping the rhizosheath size, this latter is now recognized as relevant root trait and genetic studies are being conducted to detect their QTLs and controlling genes in pearl millet, and this would provide interesting tools to breeders for the selection of efficient genotypes for a sustainable production."*

---

## Author Response (AR1)

**Manuscript number: Soil-2021-108**

Title: Rhizodeposition efficiency of pearl millet genotypes assessed on short growing period by carbon isotopes (δ13C and F14C)

**#Referee 1 comment**

**General comment**

Ndour and co-authors conducted an interesting study of rhizodeposition of pearl millet genotypes and net soil C-balance using natural 13C abundance by growing a C4 plant in a C3 soil. The findings are interesting, and the paper is generally well written. However, the authors should consider addressing the issues listed in specific comments below and some minor sentence construction/English language issues throughout the text.

> *Response: We thank #Referee 1 for his positive comments on our manuscript. We have considered the issues he listed and improved the construction of the sentences he pointed out.*

**Specific comments**

**Comment 1:** L21-22: In the abstract, provide information or explain what you mean by "low-aggregation lines" and "high-aggregation lines". Also, in L21 clarify what you mean by "was more important".

> *Response 1: We provide information in abstract to explain the mean of "low-aggregation lines" and "high-aggregation lines": "low aggregation" refers to "small rhizosheath" and "high aggregation" means "Large rhizosheath". (L23:L24)*
>
> *We clarify by replacing "the priming effect amplitude was more important" by "the priming effect amplitude ($C_{lost}/C_{new}$ ratio) was higher …. "in the abstract (L23).*

**Comment 2:** L26: "SOC" should be written in full here, followed by the acronym in parentheses.

> *Response 2: This suggestion was adopted (L28)*

**Comment 3:** L26-27: Clarify the text "increasing SOC content would have beneficial effects on agriculture by increasing soil fertility" as it implies a direct effect on soil fertility rather than indirect effect. For example, increasing SOC content would enhance soil fertility through improving physical and biological properties of the soil.

> *Response 3: The sentence was reformulated: "This will be particularly important as increasing soil organic carbon (SOC) content would enhance soil fertility through improving physical and biological properties of the soil and then would have beneficial effects on agriculture and improve food security (Lal et al., 2015) " (L28:L30)*

**Comment 4:** L27-29: "Moreover, this strategy could be particularly relevant in the Sahel region of Africa, where very little above-ground cover remains after harvest, leading to soil carbon depletion". This sentence suggests it is difficult to increase SOC in this region. Perhaps add information in this first paragraph relating to increasing/maintaining SOC in this region of Africa.

*Response 4: This information was provided in the third paragraph: (L38-L41 of the old version, and L41:L44 of the revised version): "To simultaneously improve soil carbon sequestration through crop cultivation and ensure food security in Africa, a combination of plant breeding and the development of improved agricultural practices could be considered. Then, plant breeding for root traits (architecture and root exudation) could be an original strategy to improve water and mineral nutrition of crops and to enhance SOC sequestration".*

*We think it is better in this place because the strategy we propose tackle the two problematics developed in the two first paragraphs: C sequestration in the first one and the improving pearl millet productivity in the second.*

**Comment 5:** L45: Rhizosheath is not a new trait per se.

*Response 5: We replaced "new functional trait" by "emerging functional trait" (L48)*

**Comment 6:** L53: "durability". Perhaps change this to a better term.

*Response 6: We replaced "durability of new carbon" by "fate of new carbon" to improve the sentence (L57)*

**Comment 7:** L74: "soil moistened to its water holding capacity". Do you mean 100% water holding capacity? If so, is this level of watering relevant to the Sahel region. Perhaps add information on why this level of watering was applied.

*Response 7: Yes, soil was moistened to 100% of its water holding capacity. Information was added to explain why this level of watering was applied: "This watering level was adopted to have a good germination rate as the soil was sandy with a low water retention capacity, and we have used bottomless pots which allow water infiltration as in Ndour et al (2017)"*

*This information was added in "pearl millet cultivation" sub section. (L78:L81)*

**Comment 8:** L80: "Roots and RAS were separated by washing the roots in 50 mL". Clarify whether you mean 50 mL water.

*Response 8: We have clarified by adding the missing part of the sentence: "Roots and RAS were separated by washing the roots in 50 mL **Falcon tubes containing 40 mL of distilled water**" (L86:L87)*

**Comment 9:** L83: Did you also measure root tissue $^{13}C$?

*Response 9: No, we measured $^{13}C$ only for shoot*

**Comment 10:** L91: Perhaps delete "online".

*Response 10: This suggestion was adopted (L98)*

**Comment 11:** L104-119 Section 2.4: I suggest the authors should provide more details of statistical analyses here. Currently there is insufficient details of statistical analysis (only one sentence in L118-119).

*Response 11: We have provided more information about the statistical analysis in "**Calculations and statistical analyses**" sub section: "Statistical analysis was performed using R statistical environment (version 4.0.3). The normality of the data was tested using the Shapiro test ($p < 0.05$). To test the effects of pearl-millet line on the*

*different parameters (SOC, $\delta^{13}C$, $F^{14}C$), general linear models (GLM) were constructed using the "quick linear regression" (glm) function in R. Each model was fitted by considering the distribution mode and using the corresponding link function i.e. Gaussian (link="identity") for normally distributed data and Poisson (link="log") for not normally distributed variables. Thereafter, analyses of variance (ANOVAs) were fitted to these models using the Chi square (Chisq) test, and Tukey Honest Significance Differences (TukeyHSD) post-hoc tests were performed (p < 0.05) to compare the mean of the different parameters for the four pearl millet lines using the library "multicomp" available in R." (L125:L132)*

**Comment 12:** L131: "priming effect". Perhaps here you describe net soil C-balance (i.e. negative net soil C-balance) rather than priming effect per se. Consider revising.

*Response 12: we have adopted this suggestion (L144)*

**Comment 13**: L144: "On the contrary there is no significant difference between all four bulk soils and the control soil….". Clarify whether you mean no significant difference in delta 13C values.

*Response 13: This was done: " On the contrary there is no significant difference **in** $\delta^{13}C$ between all four bulk soils and the control…" (L156)*

**Comment 14:** L161: Change "unplanted soil control" to "unplanted control soil".

*Response 14: This was done (L173)*

**Comment 15:** Fig 2C: Consider changing the unit for expressing "plant-derived C/plant biomass".

*Response 15: We remove the (%) and expressed this value as a ratio without unit*

**Comment 16:** L178: "carbon cortege". Consider an alternative term.

*Response16: We replaced "carbon cortege" by "carbon content" (L190)*

**Comment 17**: L181-182: "At this stage of plant growth, the trends showed a higher carbon loss vs gain (high Clost/Cnew ratio) for the low-aggregation and intermediary-aggregation lines L220 and L3 (4.4 and 3.5) compared to the two high-aggregation lines L132 and L253 (2.9 and 3.3) (Table 1)". This finding is interesting but there is little discussion on this in the Discussion section. Perhaps consider expanding the discussion on this.

*Response 17 : We have considered this suggestion and have added another discussion point: "Root exudates include low molecular weight molecules (sugars, amino acids, organic acids, phenolics) and high molecular weight molecules (proteins and mucilage) (Bais et al., 2006), which are probably subject to differential degradation capacity by soil microbiota. Therefore, the variation of soil C stabilization in these four pearl millet lines could be also related to a genotypic variation in the root exudates quality (biochemical composition), as reported in different plant species (Liu et al., 2019; Semchenko et al., 2021)."(L241:L245)*

*References:*

*Bais, H. P., Weir, T. L., Perry, L. G., Gilroy, S., and Vivanco, J. M.: The role of root exudates in rhizosphere interactions with plants and other organisms, Ann Rev of Plant Biol, 57, 233–266, https://doi.org/10.1146/annurev.arplant.57.032905.105159, 2006*

Liu, T.-Y., Chen, M.-X., Zhang, Y., Zhu, F.-Y., Liu, Y.-G., Tian, Y., Fernie, A. R., Ye, N., and Zhang, J.: *Comparative metabolite profiling of two switchgrass ecotypes reveals differences in drought stress responses and rhizosheath weight, Planta, https://doi.org/10.1007/s00425-019-03228-w, 2019*

*Semchenko, M., Xue, P., and Leigh, T.: Functional diversity and identity of plant genotypes regulate rhizodeposition and soil microbial activity, New Phytol, 232, 776–787, https://doi.org/10.1111/nph.17604, 2021.*

**Comment 18**: Table 1 caption: "not solvable". Do you mean the hypothesis could not be proved?

*Response 18: yes, this means that our data did not confirm this hypothesis*

**#Referee 2 (Topical Editor) comment**

**General comment**: One thorough review was obtained for your paper and unfortunately another reviewer withdrew. I have read the paper as well to provide some additional comments. In general, little is needed before this paper can be accepted for publication. Please be careful on the use of the term 'rhizosphere' when conducting measurements of rhizosheath soil. This just needs terms changed in a few places. Details of 'water holding capacity' are needed in section 2.1. Mention value and the approach used to get this measurement. Also give brief details on how the soil was moistened.

*Response: We thank you for handling our submission, and for serving as a second referee with the withdrawal of the reviewer 2.*

*We have provided in the "**Pearl millet cultivation**" subsection the value of the water holding capacity and detail of its determination: " Each pot was filled with 1.5 kg of the soil moistened to its water holding capacity (8%) which was determined gravimetrically from the graph of water loss kinetics during 18 hours. This watering level was adopted to have a good germination rate as the soil was sandy with a low water retention capacity, and we have used bottomless pots which allow water infiltration as in Ndour et al (2017)" (L78:L81)*

**Section 2.2.3.**

**Comment 1**: line 99 - 'weighed'

*Response 1: the suggested correction was done (L106)*

**Comment 2:** line 100 - 'in the pressence of…'

*Response 2: the suggested correction was done (L107)*

**Comment 3:** Line 101 'graphite was then...'

*Response 3: This suggestion was adopted (L108)*

**Section 3**

**Comment 4**: Line 145 - 'rhizosheath' rather rhizosphere

*Response 4: This suggestion was adopted (L157)*

**Comment 5**: Figure 1 - put units on the SOC % plot so i is clear it is gravimetric and not volumetric

*Response 5: This suggestion was adopted, the unit of SOC is expressed in (%wt) and this was mentioned on the Y axe title of* **Fig. 1 (a)**

**Comment 6**: line 155 - 'was found between...'

*Response 6: We replace "evidenced" by "found" (L166)*

**Comment 7**: line 158 - '(L220) and intermediary-aggregation (L3) lines...'

**Response 7**: *This suggestion was adopted* (*L169:L170*)

**Comment 8**: Fig 2 - be consistent with units. (a) and (b) should both be g kg$^{-1}$

*Response 8: In figure 2 (b) we expressed Plant-derived C mass in (g). We think the absolute value is more informative than a relative value, here.*

**Comment 9**: Fig. 2 caption - 'deposited' rather than 'deposed'

*Response 9: This suggestion was adopted (L184)*

**Comment 10**: Line 178 - '..hypothesis 3 (priming)'

*Response 10: We adopted this suggestion: only hypothesis 3 (**priming**) made it possible to solve the equations (L190)*

**Comment 11**: Future research needs/outlook is missing at present. Secondary impacts to improved soil properties would also benefit from being described

*Response 11: We added the research perspectives in the conclusion: "Further studies should focus on studying the root exudates profile among these pearl millet genotypes to see how it might help explain the observed variability in the dynamic of plant-derived C in the rhizosphere. Nevertheless, whatever the relative contribution of the quantity and the quality of root exudation in shaping the rhizosheath size, the latter is now recognized as a relevant root trait and genetic studies are being carried out to detect QTLs and to identify its controlling genes in pearl millet and this would provide interesting tools to breeders for the selection of efficient genotypes for a sustainable production." (L252:L257)*

---

## Author Response (AR2)

**Manuscript number: Soil-2021-108**

**Title: Rhizodeposition efficiency of pearl millet genotypes assessed on short growing period by carbon isotopes (δ13C and F14C)**

Topical Editor Comments to the author:

Thank you for addressing all of the comments from the reviewer and myself. All of these have addressed adequately, and the paper was prepared to a high standard from the outset.

> ***Response****: Thank you again for reviewing our paper and your positive feedback*

The 8% gravimetric water content, even for a poor moisture retaining soil, still seems very low. Check that you have not based this on air-dry weight, where soil will still retain some water.

> ***Response****: The soil WHC is low but we confirm this value and the texture characteristics we added could explain partially it*

More details of the soil used are needed. Please include texture and the FAO/WRB classification. This only needs 1 sentence and I can approve immediately once this is provided.

> ***Response:*** *We added the soil texture characteristics and its classification according to the FAO/WRB* "**It is classified as Arenosol according to the FAO/WRB (IUSS Working Group WRB, 2015) and the texture characteristics were 93% sand, 5.2% silt and 2.6% clay**" L76-L77